# Lethal Outcome of Leptospirosis in Southern Russia: Characterization of *Leptospira Interrogans* Isolated from a Deсeased Teenager

**DOI:** 10.3390/ijerph17124238

**Published:** 2020-06-14

**Authors:** Andrei E. Samoilov, Nataliya A. Stoyanova, Nikolai K. Tokarevich, Birgitta Evengard, Elena V. Zueva, Yulia A. Panferova, Yulia V. Ostankova, Elena B. Zueva, Diana E. Valutite, Evgeniy V. Kovalev, Anna R. Litovko, Alexey U. Goncharov, Alexandr V. Semenov, Kamil Khafizov, Vladimir G. Dedkov

**Affiliations:** 1Pasteur Institute, Federal Service on Consumers’ Rights Protection and Human Well-Being Surveillance, 197101 Saint-Petersburg, Russia; andrei.samoilov@gmail.com (A.E.S.); nat.stojanova@yandex.ru (N.A.S.); zoonoses@mail.ru (N.K.T.); elenazueva9@gmail.com (E.V.Z.); ersvart@inbox.ru (Y.A.P.); shenna1@yandex.ru (Y.V.O.); ezueva75@mail.ru (E.B.Z.); dianavalutite008@gmail.com (D.E.V.); alexvsemenov@gmail.com (A.V.S.); 2Central Research Institute for Epidemiology, Federal Service on Consumers’ Rights Protection and Human Well-Being Surveillance, 111123 Moscow, Russia; kkhafizov@gmail.com; 3Department of Clinical Microbiology, Umeå University, 90187 Umeå, Sweden; birgitta.evengard@umu.se; 4Department of Federal Service on Consumers’ Rights Protection and Human Well-Being Surveillance in the Rostov Region, 344019 Rostov-on-Don, Russia; kovalev@rpndon.ru; 5Center for Hygiene and Epidemiology in the Rostov Region, Federal Service on Consumers’ Rights Protection and Human Well-Being Surveillance, 344029 Rostov-on-Don, Russia; litovko.anna@yandex.ru (A.R.L.); goncharov.alyu@yandex.ru (A.U.G.); 6Martsinovsky Institute of Medical Parasitology, Tropical and Vector Borne Diseases, Sechenov First Moscow State Medical University, 119435 Moscow, Russia

**Keywords:** Leptospira, MALDI-TOF MS, PCR, high-throughput sequencing, MAT, Russia

## Abstract

This article describes a lethal case of leptospirosis that occurred in Southern Russia. The *Leptospira* strain was isolated and characterized using a microscopic agglutination test, MALDI-TOF mass spectrometry, targeted PCR, and high-throughput sequencing. We show that molecular and mass-spectrometry methods can be an alternative to conventional methods of leptospirosis diagnostics and *Leptospira* study, which require highly qualified staff and can be performed only at specialized laboratories. We also report the first whole genome of *L*. *interrogans* isolated in Russia.

## 1. Introduction

Leptospirosis is an acute zoonotic disease caused by a pathogenic spirochete of the genus *Leptospira* [1]. It is a widespread infection that affects domestic and wild animals, as well as humans [2]. Leptospirosis is estimated to infect more than a million people with an annual mortality of 60,000 cases [3,4]. Such high incidence and high mortality make it one of the leading zoonotic infectious diseases. Researchers from different countries have noted a significant underdiagnosis of leptospirosis. Costa et al., in 2015 argued that the real incidence of leptospirosis is six times higher than what is officially reported [4]. One study estimates that leptospirosis accounts for 20% to 50% of fevers of unknown etiology [5]. Significant underdiagnosis of leptospirosis and underestimation of its incidence results in a lack of proper alertness of medical doctors and misdiagnoses with tragic consequences.

Global climate change may cause new threats to human health and result in serious healthcare problems [6]. It has already caused notable warming in the high-latitude regions of Russia with its distinct uptrend between the 60th and 80th degrees of latitude [7]. The intensity of those changes in Russia exceeds global trends, especially in the Russian Arctic, where this effect is even more significant than in any other part of the country [8]. Climate change in the northern part of European Russia results in a northward shift of the forest border, so that taiga can replace tundra, which leads to the expansion of the distribution area and the increase in the population of small wild mammals which serve as major reservoirs of zoonotic infections, which account for more than 70% of human infections [9], including leptospirosis.

Recent studies have demonstrated the northward progression of both epizootic and epidemic manifestations of zoonotic infections throughout Russia [10]. Examples of these changes include outbreaks of zoonotic infections, including leptospirosis, in reindeers in northern European Russia [11]. The Nordic countries, being concerned about climate warming, have developed their action plans intended to reduce climate change-associated risks, which pose a threat to population health [12].

In addition to extreme weather and climatic conditions, anthropogenic influences, such as hostilities and drastic socio-economic changes, have a significant impact on the epidemic manifestations of leptospirosis [13]. The incidence of leptospirosis in Europe was 0.2 cases per 100,000 people; however, it has drastically increased in recent years in the Netherlands mainly due to imported cases [14]. In Russia, the average five-year incidence (2007 to 2017) was 0.2 cases per 100,000 people, according to the report of the Federal Service for Surveillance on Consumer Rights Protection and Human Well-being (http://rospotrebnadzor.ru/) [15].

The highest leptospirosis incidence is consistently reported in the Southern, Central, Volga, and Northwestern Federal Districts located in the European part of Russia with long-term incidence rates being equal to 1.7, 1.0, 0.8, and 0.64 cases per 100 thousand of the population per year. Overall, about 97% of leptospirosis cases in Russia are recorded in those territories [16,17].

In this article, we report a case of severe leptospirosis with a fatal outcome that occurred in the Rostov region of the Southern Federal District of Russia, and a study of the biological profile of the leptospira strain isolated from the patient using molecular genetics and mass spectrometry.

## 2. Case Presentation

In August 2018, a previously healthy 14-year-old male teenager presented with fever, cough, and coryza, which manifested after returning from a summer camp located near Taganrog city in the Rostov district of Russia (Figure 1). The patient was symptomatically treated for fever with paracetamol. However, his symptoms persisted. By the fifth day, the patient’s fever continued and other symptoms developed, including abdominal pain, headaches, pyrosis, vomiting, diarrhea, icterus, conjunctival suffusion, and dark urine. He was admitted to a regional hospital with a preliminary diagnosis of viral or toxic hepatitis. On examination, the patient appeared febrile, tachypneic, pale, and hemodynamically stable. Abdomen and neurological findings were normal at this point, though a moderate increase in liver size was noted. Despite clinical manifestation of leptospirosis, it was not diagnosed and symptomatic treatment was started (intravenous normal saline with drotaverin 2.0% and paracetamol). On the sixth day of onset, the patient developed gastrointestinal bleeding following by hematemesis (vomiting of blood; up to 1 L) and was found to have severe disseminated intravascular coagulation. He died shortly thereafter from circulatory failure.

Data from the postmortem studies revealed acute hepatitis of unknown etiology (viral or toxic) complicated by gastrointestinal bleeding, acute post-hemorrhagic anemia, hemorrhagic shock, severe disseminated intravascular coagulation, and circulatory failure. Tissue samples (intestine, spleen, liver, lungs, and brain) were taken and sent to the virology laboratory of the Center for Hygiene and Epidemiology in Rostov for PCR analysis to identify the pathogens of bacterial and viral etiology.

Intestinal tissues were negative for Shigella spp., enteroinvasive Escherichia coli (EIEC), Salmonella spp., thermophilic Campylobacter spp., group F adenoviruses and group A rotaviruses, norovirus genotype 2, and astroviruses using All-screen-Fl kit (B-45, AmpliSens, Moscow, Russia) as well as for enteroviruses using Enterovirus-Fl kit (R-V16-F, AmpliSens, Moscow, Russia). Lung tissue was negative for human respiratory syncytial virus (HRSV), human metapneumovirus (HMPV), human parainfluenza virus-1-4 (HPIV), ОС43, Е229, NL63 and HKUI human coronaviruses (HCoV), human rhinovirus (HRV), human B, C and E adenoviruses (HAdV) and human bocavirus (HBoV) using the ARVI-screen-FRT kit (R-V57-100, AmpliSens, Moscow, Russia). Brain tissue was negative for enteroviruses using the Enterovirus-Fl kit (R-V16-F, AmpliSens, Moscow, Russia). Liver tissue tested negative for hepatitis C virus (HCV), hepatitis B virus (HBV), and hepatitis A virus (HAV) using the HCV/HBV/HIV-FRT kit and HAV-FRT kit (RV-50-CE and R-V4-CE, AmpliSens, Moscow, Russia). All diagnostic tests were performed according to the manufacturers’ instructions. Lung tissue samples were sent to Saint-Petersburg Pasteur Institute (Saint-Petersburg, Russia) for further laboratory diagnosis.

The study has been evaluated and approved by the local Ethics Committees of the Pasteur Institute, Saint-Petersburg, Russia.

Representatives of the deceased have provided informed consent to the publication of the results of the study.

## 3. Materials and Methods

### 3.1. Leptospira Isolation

Leptospira from lung tissue was propagated at 28 °С in a liquid medium containing 80% H_2_O, 10% PBS and 10% inactivated rabbit serum (R4505-500ML, Sigma-Aldrich, USA) under control of dark-field microscopy. Second passaging was done on the eighth day and a final cell density of 107 cells per milliliter was attained on the thirteenth day of culture. The resulting cell density was suitable for serological study, mass spectrometry, and high-throughput sequencing.

### 3.2. Microscopic Agglutination Test (MAT)

MAT was used to determine the serogroup of the isolate using agglutinating sera (Armavir Biofactory, Armavir, Russia) containing antibodies against *Leptospira* of the following serological groups: *Icterohaemorrhagiae, Javanica, Canicola, Autumnalis, Australis, Pomona, Grippotyphosa, Sejroe, Bataviae*, and *Tarassovi.*

### 3.3. Molecular Detection of Leptospira

Total DNA was extracted from lung tissue using the QiaAmp DNA mini kit (Qiagen, Germany) according to the manufacturer’s instructions. Leptospira was detected using fragments of the major outer membrane lipoprotein gene (LipL32). PCR was performed to amplify a specific region of LipL32 using primers LipL32F (ACTCTTTGCAAGCATTACCGC) and LipL32R (АGCAGACCAACAGATGCAACG) [18]. DNA from L. interrogans strain RGA (Saint-Petersburg Pasteur Institute collection) was used as a positive control and deionized water was used as a negative control.

PCR reactions were carried out on a Veriti Thermal Cycler (ThermoFisher Scientific, USA), and amplicons were monitored by gel electrophoresis. The final PCR product was purified using a MiniElute gel extraction kit (Qiagen, Germany), sequenced by the Sanger method on an ABI-Prism 3500 XL sequencer (Applied Biosystems, USA) and analyzed using BLAST NCBI.

### 3.4. Matrix-Assisted Laser Desorption Ionization-Time of Flight Mass Spectrometry (MALDI-TOF MS)

Cell samples of the studied isolate were obtained by ethanol-formic acid extraction. The spectra were obtained in a linear mode operation of a Microflex LRF mass spectrometer (Bruker Daltonics GmbH, Germany) in the range m/z = 2–20 kDa using α-cyano-4-hydroxycinnamic acid as a matrix. Identification of the isolate was performed by comparison of the obtained mass spectrum with the main spectral profiles (MSP) of ten reference Leptospira strains for MAT (from Saint-Petersburg Pasteur Institute collection). The in-house MSP database was created previously [19] and exported to the main taxonomic library of reference spectra of software “Biotyper 3.1” (Bruker Daltonics GmbH, Germany). Identification was based on coincidence coefficients of the spectral peaks isolate and reference strains (logarithm of the total value of the coinciding peaks). Previously, we proposed a coefficient score value > 2.1 as the criteria to identify Leptospira strains at the serovar level [19].

### 3.5. Library Preparation and Sequencing

A library for high-throughput sequencing was prepared from 1 μg of genomic DNA using Nextera DNA sample prep kit (Illumina; San Diego, CA, USA) and sequenced on the MiSeq platform using reagent kit version 2 (Illumina; San Diego, CA, USA).

### 3.6. High-Throughput Sequencing Data Analysis

Raw reads which did not map to the human genome assembly hg38 were trimmed with Trimmomatic [20] using parameters HEADCROP:20, SLIDINGWINDOW:4:25, MINLEN:40 and CROP:127. Genome assembly was performed with SPAdes [21] using k-mer sizes of 21 and 33 nucleotides.

For phylogenetic analysis, we annotated all of the available finished L. interrogans assemblies with Prokka [22], identified core genes with Roary [23], and generated a phylogenetic tree with Fasttree using obtained alignment [24]. Read mapping and variant calling were performed with Samtools [25] and Bcftools [26] using default parameters. We have confirmed all the variant calls manually.

## 4. Results

The pathogen was isolated, propagated, and identified as L. interrogans serogroup Icterohaemorrhagiae using MAT and named Taganrog-2018.

The protein spectrum of Taganrog-2018, obtained using MALDI-TOF MS, was compared with the MSP of the reference strains of Leptospira spp. (Table 1). The highest score of coincidence (2.152) was obtained for MSP of L. interrogans serovar Copenhageni st. M20, which indicated that the isolate could be classified as serogroup Icterohaemorrhagiae and possibly to the serovar Copenhageni. However, the score value for MSP of nonpathogenic L. biflexa serovar Patoc (2.149) was very close.

As shown in Figure 2, the difference between the two MALDI spectra was mainly due to the difference in the intensity of identical peaks (Figure 2A). The samples’ spectra formed clusters of peaks in the molecular weight range m/z = 3000–3500 Da (Figure 2B). Molecular weights of the cluster maximums were m/z = 3209.5 Da for the isolate, and m/z = 3301.7 Da for the Patoc serovar, i.e., compared samples had a relative cluster peak shift by 92 Da.

The result of genotyping using the sequence of LipL32 indicated that the isolate belonged to pathogenic *Leptospira*.

This Whole Genome Shotgun project has been deposited at DDBJ/ENA/GenBank under the accession SJDW00000000. The version described in this paper is version SJDW01000000. The bacterial genome was assembled into contigs with a total length of 4.5 Mbp, N50 value of 21911 bp, and GC content of 35%.

Phylogenetic analysis (Figure 3) revealed that *L. interrogans* strain Taganrog-2018 is placed in the same clade as *L. interrogans* serovar Copenhageni strain Fiocruz L1-130 (GCF_000007685.1) and *L. interrogans* serovar Copenhageni strain FDAARGOS_203 (GCF_002073495.2), two independent assemblies of the same strain Fiocruz L1-130. We chose the latter assembly for variant calling because it had a slightly higher average nucleotide identity compared to our strain. The reference sequences were 4.63 Mbp long, which meant that our assembly covered approximately 97% of the reference genome. Variant calling revealed that *L. interrogans* strain Taganrog-2018 had 18 SNPs and 8 indels with high quality (QUAL > 35) compared to the reference genome. Genetic variations found in strain Taganrog-2018 relative to the reference are available as Appendix A (vcf file format).

## 5. Discussion

The problems of leptospirosis diagnostics are widely recognized. Standard laboratory diagnostic methods, including the cultivation of *Leptospira* and microscopic agglutination tests (MAT), are time-consuming procedures that require skillful staff with sufficient experience [27]. Proper laboratory testing for leptospirosis is not performed in almost 50% of suspected cases [4] because of the complexity of conventional diagnostic tests [5], and the need for quick, cheap, and accurate diagnostic tools is underlined in many studies [28,29,30,31,32,33,34,35]. These problems are also present in southern regions of Russia, where geographical and climatic features provide favorable conditions for the circulation of leptospirosis and create a problematic epidemiological situation [36]. Nevertheless, the subject of leptospirosis in Russia is severely under-reported in the scientific literature because of two major reasons. Firstly, most of the articles on this topic are written in Russian, and sometimes they are not available online. Secondly, almost all of the authors utilize classic microbiological methods rarely use any sequencing technology. The only exception is the paper by Voronina et al. [37], where authors have used an MLST scheme to determine the most frequent *Leptospira* sequence types in Russia and applied 454 pyrosequencing to perform whole-genome sequencing. However, most of the strains from the investigated collection were isolated in other countries and authors did not report any whole-genome assembly of *Leptospira* strains isolated in Russia.

In the current study, we used various typing methods for identification of *L. interrogans* strain Taganrog-2018, including MAT, MALDI-TOF MS, genotyping (using *LipL*32 sequence), and whole-genome sequencing. All of the methods used have shown concordant results. Molecular typing methods showed the highest resolution, with similar results using fragment analysis and high-throughput sequencing. We have established that the strain Taganrog-2018 belongs to serovar Copenhageni within Icterohaemorrhagiae serogroup of *L. interrogans.* We suggest that *LipL*32-targeted PCR can be used as an alternative to classic microbiological methods for detecting pathogenic *Leptospira* spp. Our results demonstrate the possibility for *Leptospira* detection using PCR, which has been reported previously by Barocchi et al. and Jose et al. [38,39].

We have applied MALDI-TOF mass spectrometry using the in-house library of *Leptospira* strains reference spectra for the identification of the isolate Taganrog-2018 and classified its serovar as Copenhageni. However, we have observed a high score of coincidence between the spectra of our isolate and the serovar Patoc. This can be explained by the presence of the identical peaks of the conserved ribosomal proteins in their spectra. At the same time, analysis of the spectral range of molecular weights m/z = 3000–3500 Da revealed the presence of a bell-shaped cluster of peaks, which is typical for linear polysaccharides upon the cleavage of their monosaccharide units. The difference equal to m/z = 92 Da between cluster maximums of the isolate and the sv. Patoc can be attributed to the molecular weights of three carbon atoms, three hydrogen atoms, and their hydroxyls in the monosaccharide molecules. Thus, the use of the score of coincidence alone did not allow us to identify the serovar of the isolate unambiguously. Nevertheless, manual analysis of the polysaccharide cluster peaks, which had formed in the mass spectra of leptospira in the low molecular weight range, made it possible to distinguish between *Leptospira* serovars, which have epidemiological importance because they are mostly associated with a certain type of host and thus can be used to point out possible infection reservoirs. We show that MALDI-TOF mass spectrometry can potentially be used for this purpose.

Phylogenetic analysis revealed close relations between *L. interrogans* strain Taganrog-2018 and *L. interrogans* Fiocruz L1-130, which are presented in GenBank as two independent finished genome assemblies (GCF_000007685.1 and GCF_002073495.2). The latter strain, which was isolated from a patient during a leptospirosis outbreak in 1996 [38], is widespread in Brazil. In recent work, Jaeger et al. [40] performed molecular identification of several *L. interrogans* strains isolated for over a decade from rats and dogs by sequencing the housekeeping *secY* gene, earlier suggested for *Leptospira* typing due to its variability [41], and found that all sequences were 100% identical among themselves and to the strain *L. interrogans* Fiocruz L1-130. *L. interrogans* strain Taganrog-2018 has one SNP in the *secY* gene (Appendix A), which makes it distinguishable from strains isolated from Brazilian rats and dogs, but limitations of our study do not allow us to draw any substantial conclusions about the occurrence of *Leptospira* species: to our knowledge, we are presenting the first full-genome assembly of *Leptospira interrogans* isolated in Russia. Further research using the latest NGS technologies will allow us to gain insights into phylogenetics and determine the major clonal populations of *Leptospira* circulating in Russia.

## 6. Conclusions

Leptospirosis is one of the most common zoonoses in the world. The incidence of leptospirosis in Russia is sporadic and physicians should consider this infection during differential diagnosis of patients with fevers, especially in the warm season. In the described case a diagnostic error was fatal. We have isolated a pathogenic strain of *L. interrogans* serovar Copenhageni and used several diagnostic and research methods to study it. Conventional classic methods are complicated and require the cultivation of a large number of *Leptospira* serovars and skillful staff with sufficient experience. We propose a LipL32-targeted PCR reaction as a method which will increase the speed and quality of laboratory diagnostics of leptospirosis and mass spectrometry analysis based on the determination of the leptospiral O-antigens oligosaccharide subunits as a method which will allow for the identification of bacterial serovar and, consequently, the likely source of infection.

We have also performed whole-genome sequencing of the obtained strain, which allowed us to report the first full-genome assembly of *L. interrogans* isolated in Russia.

## Figures and Tables

**Figure 1 ijerph-17-04238-f001:**
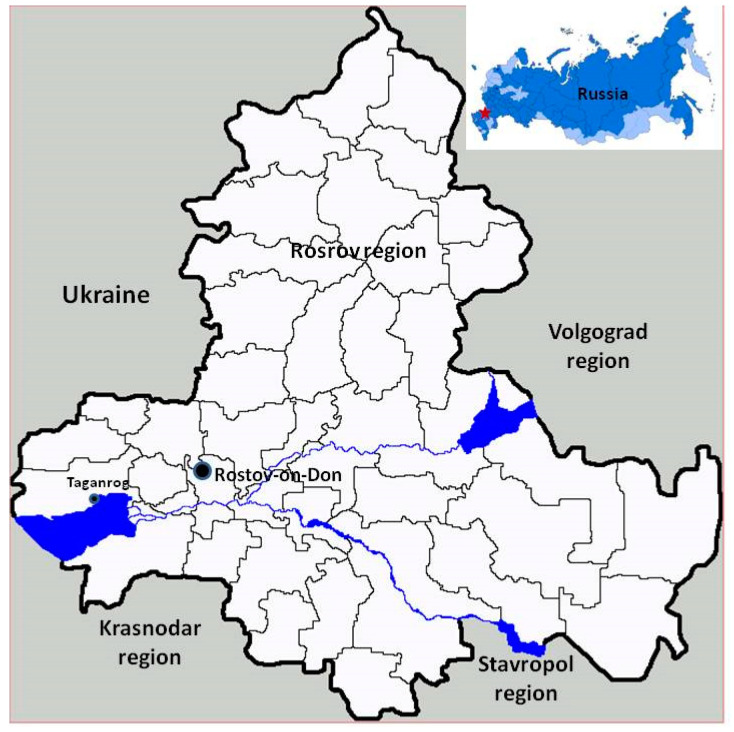
Location of Taganrog city in Russia (47°14′10″ N, 38°53′48″ E.).

**Figure 2 ijerph-17-04238-f002:**
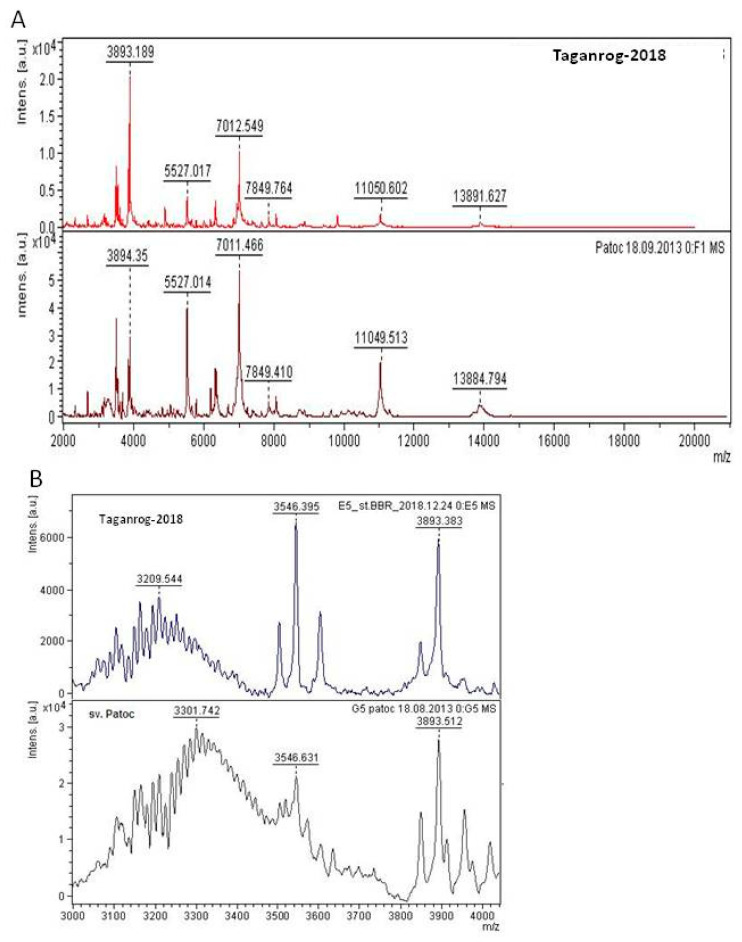
MALDI-TOF spectra of the studied isolate Taganrog-2018 and the reference strain of *L. biflexa* Patoc I serovar Patoc. (**A**) Mass peaks in the molecular weight range m/z = 2000 − 20000 Da; (**B**) Cluster of peaks in the range m/z=3000-3500 Da.

**Figure 3 ijerph-17-04238-f003:**
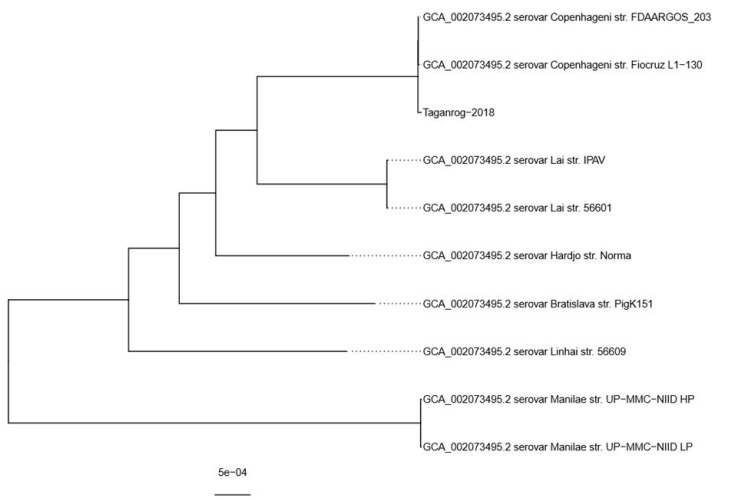
A midpoint-rooted phylogenetic tree of all available finished *L. interrogans* genomes and our assembly. Serovar and strain names are provided at the tip of labels.

**Table 1 ijerph-17-04238-t001:** The spectral coincidence of the studied *Leptospira* strain with the spectral profiles of 10 reference strains of *Leptospira* spp. represented in the project database MALDI Biotyper 3.1.

N	Species	Score	Similarity
1	*L. interrogans* sv. Copenhageni, st.M 20	2.152	high
2	*L. biflexa* sv. Patoc, st. Patoc I L. bifl	2.149	high
3	*L. interrogans* sv. Icterohaemorrhagiae, st. RGA	2.123	high
4	*L. interrogans* sv. Mozdoc st. 5621	2.080	high
5	*L. interrogans* sv. Pomona, st. Pomona	1.972	middle
6	*L. interrogans* sv. Seiroe st. M84	1.885	middle
7	*L. interrogans* sv. Grippotyphosa, st. Moscow V	1.850	middle
8	*L. interrogans* sv. Saxkoebing st. Mus24	1.817	middle
9	*L. interrogans* sv. Canicola st. Hond Utrecht IV	1.628	low
10	*L. interrogans* sv. Tarassovi st. Perepelitsin	0.423	low

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
