# Peer review of "Lethal Outcome of Leptospirosis in Southern Russia: Characterization of Leptospira Interrogans Isolated from a Deceased Teenager"

_ijerph, 2020, doi:10.3390/ijerph17124238_

Round 1

Reviewer 1 Report

Well-written case report with great amount of detail in the background and throughout to make this a nice reference paper for those considering a similar diagnosis.

I would note that I think the level of detail in the methods may be a little much. I do not work as a medical microbiologist, but I recognize many of the extraction protocols as standard. I would try to summarize those aspects which have deviated from standard/typical approach.

Otherwise, very minor edits suggested in attached pdf. Interesting and concerning case.

Author Response

Dear colleague,

thank you for your review and corrections. We have applied all of the edits you have suggested.

“I think the level of detail in the methods may be a little much. I do not work as a medical microbiologist, but I recognize many of the extraction protocols as standard. “ - we have removed excessive descriptions of standard laboratory procedures

Best regards

Reviewer 2 Report

Sir, 

first at all thank your for the opportunity to review this very interesting paper on a deadly case of Leptospirosis form the Rostov Oblast. Authors report a case that may be of significant interest for the international reader for the following reasons:

1) the Rostov Oblast (i.e. the Rostov region) is endemic for severe cases of leptospirosis, in both animals and humans (e.g. https://pubmed.ncbi.nlm.nih.gov/2970742/; https://pubmed.ncbi.nlm.nih.gov/13423820/; https://pubmed.ncbi.nlm.nih.gov/2955616/) but, as the aforementioned reports were published only in Russian, well before 1991, international reader may fail to understand the novelty of a report suggesting an epidemiological link between a case of leptospirosis occurring in Southern Russia and a leptospira subgroup that was previously found epidemic in Brazil, and nearly 20 years before the index case!

2) the Authors have suggested that the lab probes may be significantly biased, with subsequent association of their case to a strain (i.e. Copenhageni) that was presumptively unrelated with the actual pathogen, if I correctly understood the results. If it is correct, this is a significant warning for all lab professionals around the world, who may incorrectly associate pathogens with their gene library, with subsequent failure in epidemiological studies.

3) the case is also clinically of absolute interest, as it stresses how early symptoms of leptospirosis may fail to catch up an appropriate diagnosis, delaying the etiological therapies and ultimately impairing the final diagnosis of the patient.

For all the aforementioned reason, I think that this paper may deserve a full publication on IJERPH.

HOWEVER, I also think that the present paper will require an extensive overhaul. 

Firstly, even the English language is substantially appropriate in terms of grammar, the phrasing must be improved. Authors could be helped in their redrawing by following the subsequent suggestions:

a) shorten the introduction, focusing on the actual understanding of leptospira infection in European Russia;

b) the details on the lab exams are of absolute interest for laboratory professionals, who represent only a fraction of the readers of IJERPH. Therefore, evaluate whether a part of such section may be included as an "annex" at the end of the paper.

c) I think that points 1-2-3 represent the main messages of your paper: please discuss them giving an appropriate level of detail.

Author Response

Dear colleague,

thank you for your review. Concerning your suggestions:

“Firstly, even the English language is substantially appropriate in terms of grammar, the phrasing must be improved.” - we have revised English text in the the whole article

  1. a) “shorten the introduction, focusing on the actual understanding of leptospira infection in European Russia” - done
  2. b) “the details on the lab exams are of absolute interest for laboratory professionals, who represent only a fraction of the readers of IJERPH. Therefore, evaluate whether a part of such section may be included as an "annex" at the end of the paper.” - done, we decided to omit unnecessary details as they are mostly standard procedures
  3. c) concerning points 1 and 2:
  4. “international reader may fail to understand the novelty of a report suggesting an epidemiological link between a case of leptospirosis occurring in Southern Russia and a leptospira subgroup that was previously found epidemic in Brazil, and nearly 20 years before the index case!” - we have removed all of the excessive mentions about Brazil to avoid confusion, because connecting our case with Brazil outbreak was not our goal;
  5. “the Authors have suggested that the lab probes may be significantly biased, with subsequent association of their case to a strain (i.e. Copenhageni) that was presumptively unrelated with the actual pathogen, if I correctly understood the results” - if we understood this point correctly, you are referring to the bias in the reference genome. Whst we meant is that reference genome might be not a DNA sequence perfectly representing DNA sequence of actual Leptospira and that it probably has some sort of technical artifacts. This is expected from WGS genome assemblies, but in this case it was very distinctive because the true biological difference between our strain and the reference strain was rather small compared to technical artifacts. We decided to remove this sentences from our article because discussion of limitations of whole-genome assemblies is beyond the scope of our paper.
  6. “it stresses how early symptoms of leptospirosis may fail to catch up an appropriate diagnosis, delaying the etiological therapies and ultimately impairing the final diagnosis of the patient.” - done, we have discussed the problem of leptospirosis underdiagnostics

Best regards

Reviewer 3 Report

The manuscript by Stoyanova et al. deals with the identification of a Leptospira isolated from an undiagnosed clinical case who resulted in patient death.

Authors efforts in pathogen identification were based on MAT, MALDI-TOF MS, Genome sequencing and gene sequencing. Unfortunately, results from genome sequencing were incomplete, so that many information potentially very useful and interesting got lost. Nevertheless, the two molecular approaches produced consistent results.

Investigations about the isolate were justified by the low incidence (or underdiagnosis?) of such infection in Russia and by the lack of genetic information about Russian isolates so that assessing genetic relationships might be of interest.

The manuscript, however, would need some improvements. I would encourage authors to revise the discussion section (it is mostly a summary, in the present form), as well as conclusions and abstract.

In the conclusions section, MAT suddenly reappears (it had been named only at the beginning of results, without comments or further information; if author think this technique is worth of attention, a better explanation should be provided to readers), whereas MALDI-TOF is missing (though I think that proposing it as a diagnostic tool might be not so appropriate and useful, as instead stated in L246-248). Moreover, the authors conclude that a method based on qPCR should be developed. In this regard, I'd encourage the authors to do a deeper literature search (where PCR, qPCR as well as LAMP tests are reported), to refer to such data in proposing (or planning to develop) more rapid and effective diagnostic approaches. This might be much beneficial to conclusions and, maybe, discussion.

Moreover, authors refer to a "diagnostic error" in lines 265-266", but in lines 95-96 they claim that the pharmacological treatment had been started based on clinical features suggestive of leptospirosis. It is not clear if the infection had been hypothesized or not, although the described treatment was clearly inappropriate.

Minor points:

L199:  L. Icterohaemorrhagiae should be  L. icterohaemorrhagiae
L258: what does 37 mean? (...Brazil [38]. 37 However... )

Author Response

Dear colleague,

thank you for your review. Concerning your suggestions:

“Unfortunately, results from genome sequencing were incomplete, so that many information potentially very useful and interesting got lost.”

“I would encourage authors to revise the discussion section (it is mostly a summary, in the present form), as well as conclusions and abstract.”

“In the conclusions section, MAT suddenly reappears (it had been named only at the beginning of results, without comments or further information; if author think this technique is worth of attention, a better explanation should be provided to readers), whereas MALDI-TOF is missing” - done, now discussion includes review about problems of leptospirosis diagnostics, lack of scientific papers about leptospirosis in Russia, applicability of methods used (MALDI-TOF and MAT) and results of genome sequencing

“though I think that proposing it as a diagnostic tool might be not so appropriate and useful, as instead stated in L246-248” - we have clarified that we suggest use of MALDI-TOF as the alternative to conventional methods of Leptospira serotyping, not leptospirosis diagnosis

“Moreover, the authors conclude that a method based on qPCR should be developed. In this regard, I'd encourage the authors to do a deeper literature search (where PCR, qPCR as well as LAMP tests are reported), to refer to such data in proposing (or planning to develop) more rapid and effective diagnostic approaches. This might be much beneficial to conclusions and, maybe, discussion.” - we removed this sentence because methods which allow to detect pathogenic Leptospira are already exist and we use it (LipL32-specific PCR).

“Moreover, authors refer to a "diagnostic error" in lines 265-266", but in lines 95-96 they claim that the pharmacological treatment had been started based on clinical features suggestive of leptospirosis. It is not clear if the infection had been hypothesized or not, although the described treatment was clearly inappropriate.” - done, we have repharased those sentences, now it should be clear that, despite clinical features suggestive of leptospirosis, it was not diagnosed and that was a fatal diagnostic error.

Minor points: fixed.

Best regards

Round 2

Reviewer 2 Report

Authors have addressed all of my concerns.

Therefore, I've no further requests.

(please be aware that some format mistakes are scattered across the text as a consequence of the extensive editing, but that may be fixed in the pre-print session)

Reviewer 3 Report

The revised version shows several improvements, addressing most reviewer's requests, which makes it suitable for publication.

I'd only suggest some minor improvements in the English language throughout the text.